# Integrating In Vitro Analytics for Improved Antibody–Drug Conjugate Candidate Selection

**DOI:** 10.3390/cancers18010164

**Published:** 2026-01-03

**Authors:** Virginia del Solar, Ali Saleh, Annarita Di Tacchio, Lena Sokol Becciolini, Gyoung Dong Kang, Bianka Jackowska, Yan Hu, Chao Gong, Angel Zhang, Leigh Hostetler, Maximilliam Lee, Akbar H. Khan, Abhisek Mitra, Mahammad Ahmed, David Tickle, Balakumar Vijayakrishnan

**Affiliations:** 1Oncology Targeted Discovery-Drug CandidateDC, Chemistry, AstraZeneca, London E1 2AX, UK; ali.saleh@astrazeneca.com (A.S.); annarita.ditacchio@astrazeneca.com (A.D.T.); lena.becciolini@astrazeneca.com (L.S.B.); gyoung-dong.kang@astrazeneca.com (G.D.K.); maximillian.lee@astrazeneca.com (M.L.); mahammad.ahmed@astrazeneca.com (M.A.); david.tickle@astrazeneca.com (D.T.); balakumar.vijayakrishnan@astrazeneca.com (B.V.); 2Oncology Targeted Discovery-Drug Candidate, Chemistry, AstraZeneca, London E1 2AX, UK; biankagjackowska@gmail.com; 3Oncology Targeted Discovery-Drug Candidate Bioscience Research and Discovery, Translational Pharmacokinetics/Pharmacodynamics, AstraZeneca, Waltham, MA 02451, USA; yan.hu6@astrazeneca.com; 4Oncology Targeted Discovery-Drug Candidate Bioscience Research and Discovery, Integrated Bioanalysis, AstraZeneca, Gaithersburg, MD 20878, USA; chao.gong1@astrazeneca.com; 5Oncology Targeted Discovery-Drug Candidate Bioscience Research and Discovery, AstraZeneca, Gaithersburg, MD 20878, USA; angel.zhang@astrazeneca.com (A.Z.); leigh.hostetler@astrazeneca.com (L.H.); abhisek.mitra@astrazeneca.com (A.M.); 6Oncology Targeted Discovery-Drug Candidate, Chemistry, AstraZeneca, Waltham, MA 02451, USA; akbar.khan@astrazeneca.com

**Keywords:** Antibody–drug conjugates, enzymatic assay, ADC serum stability, computational integration

## Abstract

Antibody–drug conjugates (ADCs) are targeted antitumoral medicines that bind potent drugs to antibodies, directing treatment to cancer cells and reducing side effects. In early drug development, finding the optimal ADC candidates was scientifically and technically challenging since several critical factors must be considered, such as ensuring these drugs remain stable in the plasma during circulation or confirming they can release their payload where it is needed. Herein, we combine laboratory-based analytical workflows with advanced data analysis pipelines to quickly assess the stability and efficacy of ADC candidates. Our approach helps in selecting the promising candidates for further development by identifying the transformations an ADC undergoes in serum and the efficacy on releasing their payloads. This integrated method allows us to analyse a larger set of compounds efficiently and supports the discovery of safer, more effective cancer treatments.

## 1. Introduction

Antibody–drug conjugates (ADCs) represent a major advance in targeted oncologic therapeutics, combining the specificity of monoclonal antibodies with the potent activity of cytotoxic drugs to achieve selective tumour cell killing whilst minimising off-target toxicities [1,2,3]. However, translating this strategy into successful clinical treatments has proven complex, as it involves several design and development challenges. Despite extensive research over recent decades, only few ADCs have received clinical approval. Recent innovations in the chemistry of payloads and linkers, along with antibody engineering, have substantially improved the safety and efficacy profiles of ADCs [4]. Nevertheless, achieving effective treatment for challenging types of cancer, such as triple-negative breast cancer and various solid tumours, remains an obstacle [5,6]. The approval of ADCs such as trastuzumab deruxtecan highlights progress in the field, while ongoing investigations continue to explore new targets and strategies to advance clinical outcomes for oncology patients.

Structurally, an ADC consists of a monoclonal antibody targeting a cell surface antigen, a cytotoxic payload, and a chemical linker, each contributing uniquely to pharmacology, efficacy, and safety [7]. However, the clinical success of ADCs depends on optimising and coordinating those three elements. Differences in antibody affinity, drug potency, and the stability of the chemical linker can significantly impact how stable the ADC is over time and how it is processed in the body, as well as influence immunogenicity and aggregation [8,9,10,11,12,13]. Linker choice, cleavable or non-cleavable, deeply impacts plasma stability and intracellular payload release, making molecular design a crucial determinant of the ADC therapeutic effect [14,15,16,17]. These key features need robust analytical methodologies throughout the development process. Advanced in vitro techniques such as mass spectrometry, hydrophobic interaction chromatography, and ligand binding assays enable accurate determination of DAR, ADC heterogeneity, payload release kinetics, and potential metabolism [18,19,20,21]. These analytics provide essential insights into ADC stability and performance, enabling potentially predictive modelling of pharmacokinetics, pharmacodynamics, and immunogenicity profiles in vivo [22,23,24,25].

The continued success and expanding clinical use of antibody–drug conjugates highlight the importance of accurately optimising each ADC component to ensure therapeutic effectiveness and manage toxicity [26]. Emerging advances in combination therapies, molecular imaging, and analytical platforms have enhanced our ability to assess target expression, biodistribution, and pharmacodynamic effects, thus streamlining candidate selection and improving outcomes in both preclinical and clinical trials [27]. As ADC design continues progressing, approaches such as strategic target selection, improved conjugation technologies, and real-time data analysis will be central in creating safer and more effective antineoplastic drugs [28].

As bioanalytical workflows have evolved, their integration has become a key factor in ADC candidate selection, enabling the identification of molecules with optimal stability, efficacy, and safety for clinical advancement [29,30]. This article discusses the analytics mainly used by our analytical team, guiding the selection of candidates with optimal in vivo profiles. Although a variety of advanced workflows are available, this study focused on assays that we consider key for predicting in vitro potency and in vivo stability, specifically, enzymatic assays and the evaluation of ADC stability in serum. In addition, we show the agreement between the bioanalytical results and the biological data, including in vitro cytotoxicity and in vivo pharmacokinetics. Finally, we comment on how integrating AI tools as Power BI into the analysis pipeline reduces the turnarounds and variation between operators.

In summary, these targeted analyses provide detailed insight into payload release, biotransformation, and stability, and highlight how integrated in vitro analytics can improve candidate selection and prediction of in vivo outcomes.

## 2. Materials and Methods

### 2.1. Bioconjugation of ADCs

ADC candidates were generated by conjugating cytotoxic payloads to the antibody backbone via specified linkers using established bioconjugation protocols. The resulting ADCs were purified by standard chromatographic techniques and characterised for monomer purity by SEC-HPLC and drug-to-antibody ratio (DAR) by RP-HPLC and LC-MS (Appendix A).

Briefly, both linker-payloads were added as a DMSO solution (16 molar equivalent/antibody, 0.64 µmol, in 0.13 mL of DMSO) to trastuzumab antibody solution in PBS, 1 mM EDTA, pH 7.4 (6.0 mg, 40 nmol) for a 10% (*v*/*v*) final DMSO concentration. For bromoacetamide linker conjugation, 15% of borate buffer, pH 8.5, was added to obtain a final pH of 8.3. Both linker solutions were incubated at room temperature for 1 h with gentle shaking and quenched by addition of *N*-acetyl cysteine (3.2 µmol, 32.0 mL at 100 mM), followed by purification in PBS pH 7.4 using AKTA-SECPrep (HiLoad^®^ 26/600 Superdex^®^ 200 pg, GE28-9893-36, Cytiva, Marlborough, MA, USA) and formulated in 20 mM Histidine/Histidine HCl and 240 mM sucrose, pH 6.0, by spin filtration using a 15 mL Amicon Ultracell 30 kDa MWCO spin filter (UFC5030, Millipore, Burlington, MA, USA), sterile-filtered, and analysed.

### 2.2. B-Glucuronidase Assays

Beta-glucuronidase (5 ng/µL, prepared from a 0.5 µg/µL stock in 50 mM sodium acetate, pH 5.0; 6144-GH-020, R&D Systems, Minneapolis, MN, USA) was incubated at 37 °C with the linker-payload (0.5 mM, prepared by diluting a 10 mM DMSO stock), for a total volume of 200 µL. At 0, 2, 4, and 24 h, 30 µL aliquots were removed and quenched with an equal volume of reversed-phase buffer. Payload release was quantified by reversed-phase HPLC (RP-HPLC) and results were extrapolated into the corresponding calibration curves for quantification (Appendix A). All reactions and analyses were performed in duplicate. Statistical analysis was performed for each time point using an unpaired two-tailed *t*-test.

### 2.3. ADC Catabolism Assay

Human liver lysosomes (H0610.L, Xenotech, Kansas City, KS, USA) were used to assess ADC catabolism in vitro. In a final volume of 300 µL, ADC was diluted to a final concentration of 0.5 µM and incubated with lysosomal extract at a final concentration of 0.1 mg/mL in 10× Catabolic buffer (103307-306, Xenotech, Kansas City, KS, USA) (pH 5.0) to simulate lysosomal conditions. DTT was added in the buffer to ensure activation of key lysosomal enzymes such as Cathepsins. Reactions were carried out at 37 °C at different time points (0, 2 h, 4 h, 24 h, 48 h). At each time point, reactions were quenched by addition of acetonitrile containing 2 µM tolbutamide at a sample/acetonitrile ratio of 1:2, followed by centrifugation at 4000 rpm for 30 min at 4 °C. Supernatants were analysed by LC–MS/MS and the cleaved payload was detected and quantified.

Payload release was normalised to the area of the internal standard in each sample. Negative controls (no lysosomal extract) and positive controls (including known cleavable ADCs under established conditions) were included in each assay. All reactions and analyses were performed in triplicates. Statistical analysis was performed for each time point using an unpaired two-tailed *t*-test.

### 2.4. ADC Serum Stability Studies

The sample underwent three different steps: serum incubation, immunoprecipitation, and, finally, LC-MS data acquisition and analysis.

#### 2.4.1. Serum Incubation

ADCs were mixed with mouse serum (MSE00SRM-9953K, BioIVT, Westbury, NY, USA) and plated in duplicates in three independent plates (t0, t1, and t2, corresponding to t = 0, t = 24 h, and t = 72 h). t1 and t2 plates were stored in an incubator (37 °C, 5% CO_2_) for 24 h and 72 h and then frozen. The t0 plate was frozen immediately after plating. QCs (samples diluted in PBS instead of serum) were plated alongside serum samples in the t0 plate.

#### 2.4.2. Immunoprecipitation

Samples were captured by the magnetic beads (88827, ThermoScientific, Waltham, MA, USA) conjugated with human IgG-derived Fc-fusion protein (7103322500, ThermoScientific, Waltham, MA, USA) at a volume ratio of 3:1 (sample/beads). After the capture, samples were washed and then eluted by incubation with elution buffer for 30 min at rt and 1200 rpm. 

#### 2.4.3. LC-MS Data Acquisition

Upon elution, the sample pH was adjusted by sodium phosphate addition followed by incubation with 0.8 µL of PNGaseF (P0704L, New England Biolabs, Ipswich, MA, USA) at 37 °C for 2 h. Upon deglycosylation, dithiothreitol (A39255, Thermofisher Scientific, Waltham, MA, USA) was added to the sample at a final concentration in the sample of 100 mM. The 96-well plate was shaken for 30 min at 1200 rpm for complete reduction. Subsequently, an equal volume (1:1 *v*/*v*) of 50% acetonitrile was added to denature the proteins, followed by injection into the LC-MS system.

For data analysis, refer to Section 2.7. Data Analysis and Visualisation.

### 2.5. In Vitro Cytotoxicity Assays

ADC cytotoxicity was evaluated by using HER2-positive SKOV3 cultured in 3D spheroids. Briefly, SKOV3 cells were seeded at a density of 3000 cells/wells in a 96-well plate. After 48 h, the ADC stock was diluted in culture medium and serially diluted nine times (5-fold each). Cells were treated in duplicate with these ADC dilutions, whilst control cells were treated only with vehicle. Plates were incubated under standard conditions (37 °C, 5% CO_2_) for 6 days. Cell viability was measured using Cell-Titer-Glo (Promega, Madison, WI, USA) per the manufacturer’s instructions, and luminescence recorded. Cell survival was calculated relative to the mean luminescence of six untreated control wells (100%). IC_50_ values were determined from dose–response curves using non-linear regression (four-parameter sigmoidal fit) in GraphPad Prism version 10.2.0 (GraphPad Software, Boston, MA, USA). The ADC controls used in the studies were the isotype ADC analogues, NIP228-WT.

### 2.6. In Vivo Pharmacokinetic Studies

#### 2.6.1. Mouse Pharmacokinetic Study

Athymic nude female mice were purchased from Charles River Laboratories and randomised based on body weight using the Pure Random method. For pharmacokinetic studies, naïve mice (aged 8–10 weeks, n = 9 per group) were administered a single intravenous injection of vehicle control or ADC. For ADC-treated groups, each animal received a single intravenous dose of 5 mg/Kg. Blood samples were collected at eight distinct time points (30 min, 2 h, 6 h, 24 h, 48 h, 4 days, 7 days, 10 days, 14 days, and 21 days post dose).

To minimise animal use, each mouse underwent two blood collections, an initial submandibular bleed followed by a terminal bleed. Blood samples were collected in K2EDTA tubes, with an in-life sample volume of less than 200 µL per mouse. Following collection, whole blood was gently mixed and stored on ice until processing. Samples were centrifuged (10 min, at 12,000 rpm and at 4 °C) to separate plasma. Plasma was then transferred to fresh tubes and stored at −80 °C until analysis

#### 2.6.2. DAR and ADC Concentrations

DAR and ADC concentrations were determined by immunoprecipitation followed by LC-MS analysis. Briefly, samples were split into two aliquots for measurements of total antibody and total ADC.

For total antibody quantification, samples were captured by the magnetic beads (88827, ThermoScientific, Waltham, MA, USA) conjugated with human IgG-derived Fc-fusion protein (7103322500, ThermoScientific, Waltham, MA, USA). After capture, samples were washed and subjected to on-bead proteolysis with trypsin. Following trypsin digestion, characteristic peptide fragments derived from the antibody were analysed using an MRM LC-MS method.

For total ADC measurement, the same bead and capture chemistry was used. However, after capture, samples were washed and underwent on-bead hydrolysis with papain. Subsequently, the released payload was analysed by the MRM LC-MS method.

### 2.7. Data Analysis and Visualisation

LC-MS raw files were deconvoluted using BioPharma Finder (BioPharma Finder 5.2, ThermoScientific, Waltham, MA, USA) to identify the masses of the species detected in each sample (Appendix A). The deconvoluted data, along with the linker-payload mass and the heavy and light chain masses for the antibody, were imported into Power BI. Automated pipelines were used for rapid data processing, report generation, and visualisation. All species assignments were manually verified before report completion.

Statistical analysis was performed only for enzymatic and catabolic studies. Cutoff criteria were used for in vitro cytotoxicity and plasma stability datasets as part of a go/no-go decision process, whilst the objective for in vivo study was to obtain a descriptive overview of PK properties of the molecule.

#### PowerBI Analysis

Preprocessing was performed using Power Query (PowerBI version 2.129.1229.0 64-bit; PowerBI, Redmond, WA, USA) to gather all the intensity values and filter them based on time points and replicas, as well as providing as input the theoretical masses for L0, H0, the linker-payload, and the possible transformations (i.e., hydrolysis, glycosylation, deconjugation). Deconjugation and hydrolysis rates and DAR values were calculated with DAX formulas. Stability trends were visualised using interactive bar charts. Reports were exported as standardised dashboards for further review.

For details on PowerBI code, please refer to SI (Appendix A, “Identification of Species or Components”, and “Calculations” sections).

### 2.8. Ethics Statement

All animal experiments were conducted in a facility accredited by the Association for Assessment and Accreditation of Laboratory Animal Care under the guidelines of AstraZeneca’s Institutional Animal Care and Use Committee and appropriate animal research approvals.

## 3. Results

### 3.1. Enzymatic Studies on Linker-Payload and ADC

ADCs are considered a powerful alternative to conventional antineoplastic agents since they selectively bind to specific antigens on the surface of cancerous cells and release a toxic payload inside the cell upon internalisation via endocytosis and transfer to lysosomes. This multi-step mechanism offers significant benefits over traditional chemotherapy, including improved efficacy and reduced side effects.

Given this mechanism of action, enzymatically cleavable linkers are favoured in early development of ADCs. This type of linker favours the release of cytotoxic payloads inside the tumour cells through specific cleavage by lysosomal enzymes. Therefore, evaluating the enzymatic cleavage of the linker-payload (LP) or investigating the ADC catabolism by lysosomes is crucial for understanding the influence on ADC pharmacokinetic and therapeutic effects.

#### 3.1.1. Enzymatic Studies on Linker-Payload

Extensive research describes the importance of enzymatic analysis in drug development [14,15,16,30,31,32,33]. Accordingly, our early assays for triaging the linker-payload cleavability are enzymatic studies. These data alongside the payload stability and potency are key criteria defining whether a molecule advances to bioconjugation or is excluded from further development. Although enzymatic assay results can offer early indications of ADC cytotoxic potential, it should be taken with precaution, as multiple factors influence the mechanism of action leading to cell death.

Two enzymatic assays were used to measure payload release after enzymatic activity. Routine analyses include β-glucuronidase and cathepsin B cleavage, quantified by RP-HPLC. To note, in cases where the linker-payload presents multiple species that cannot be identified by UV absorption, additional analysis using an LC-MS method was applied.

To illustrate herein the value of a rational analytical workflow, we selected two representative LPs containing a β-glucuronide linker (Figure 1A) and assessed their integrity by β-glucuronidase enzymatic assay (Figure 1B).

Enzymatic results showed that, after 24 h, the maleimide and bromoacetamide linker-payloads released 46% and 60% of their cytotoxic payload, respectively. This suggests that both LPs could exhibit cell death in a biological environment, despite the fact that a substantial amount of the molecule remains intact after 24 h. However, it is important to bear in mind that a higher rate of enzymatic cleavage does not necessarily translate to increased cytotoxicity.

Internal analyses of different LP panels have revealed no consistent correlation between susceptibility to enzymatic cleavage and ADC potency, which can be attributed to differences in the payload activity. Therefore, while enzymatic assays are valuable for predicting cytotoxic potential, multiple factors contribute to the final cytotoxic profile, and candidate triage should not be based on one study alone.

#### 3.1.2. Enzymatic Studies on ADC

After internalisation via endocytosis, ADCs are trafficked to lysosomes, where proteolytic cleavage releases the cytotoxic payload, leading to cell death. It is essential to determine whether ADCs effectively release active payloads that can initiate their intended cytotoxic effects. Following conjugation, ADC stability can be studied in the presence of the lysosomal extract, providing valuable insight into both the kinetics of intracellular payload release and the subsequent cytotoxic effect. This type of study would estimate the efficacy of the ADC in in vivo studies alongside other enzymatic studies and the cytotoxic data.

Briefly, the ADC catabolism assay by lysosomes is a complex workflow involving ADC incubation with lysosomal extract over a specified time course, followed by protein precipitation, centrifugation to remove insoluble material, and subsequent LC-MS analysis for targeted detection of the released payload.

Catabolic results showed that, after 24 h and 48 h of incubation, Her-maleimide and Her-bromoacetamide ADCs released comparable amounts of Exatecan after incubation in lysosomal extract (Figure 1C). Similarly to the enzymatic assay results, Her2-bromoacetamide exhibited a slightly higher payload release; however, the difference between the two ADCs was not statistically significant. This finding indicates that both ADCs can efficiently release their cytotoxic payload under intracellular conditions. Notably, the results are consistent with the enzymatic study data, further supporting the expected in vitro efficacy of both ADCs.

This approach enables the identification of linkers that efficiently release their payload in the presence of lysosomal enzymes. Additionally, modifications on payloads in the lysosomal extract can be evaluated, as these may affect the potency. Such information is highly relevant for evaluating in vitro efficacy and predicting the mechanism of action of ADC candidates.

#### 3.1.3. In Vitro Evaluation of Cytotoxic Activity

To assess the biological activity of ADCs containing these LPs, subsequent in vitro cytotoxicity studies are required. Thus, the Her-maleimide and Her-bromoacetamide ADCs were tested against the HER2-overexpressing SKOV3 cell line—a cell line notably resistant to some cytotoxic reagents [34,35]. To better mimic in vivo conditions and to enable a more precise evaluation of the drug efficacy, we performed a 3D spheroid culture model. For rigorous data validation, isotypic ADC analogues were included in the treatment as controls.

The dose–response curves depicted in Figure 1D showed that changing the conjugation handle did not impact the linker-payload activity, as Her-maleimide and Her-bromoacetamide ADCs exhibited comparable cytotoxicities. In addition, the isotype ADCs exerted significantly lower cytotoxicity, probing the selectivity of the ADCs of interest.

These results were in accordance with the enzymatic data and the lysosomal payload release results where the bromoacetamide payload was slightly more cleaved than the maleimide one, implying that more Exatecan is released inside the cell. This hypothetically higher concentration of Exatecan inside the cell could explain the lower IC_50_ of Her-bromoacetamide with respect to the one of Her-maleimide. Hence, combining the enzymatic evaluation of LPs with ADC catabolic studies in lysosomal extract might be a necessary step to accurately assess the potential efficacy of ADC candidates.

### 3.2. In Vitro and In Vivo ADC Studies

#### 3.2.1. ADC Serum Stability Results and In Vivo PK Data

ADC stability is a Critical Quality Attribute (CQA) which might predict the in vivo pharmacokinetic outcomes. For instance, it is well known that the stability of an ADC in mouse serum might be compromised due to the presence of carboxylesterase 1C, impairing the preclinical evaluation [24]. Ideally, a promising ADC candidate would present high serum stability, whilst it would simultaneously ensure efficient lysosomal payload release. In addition, understanding the biotransformations suffered during serum incubation may give valuable insights regarding both therapeutic efficacy and the potential off-target cytotoxicity.

To assess these challenges, our workflow involves immunoprecipitation (IP), to reduce the matrix effect, followed by LC-MS analysis and PowerBI-based data processing upon deconvolution. This approach has been designed to selective analyse the ADC and its derivatives rather than tracking the small molecules released during incubation. By aiming for the ADC and its transformations, we accelerate the data interpretation and bypass the time-consuming identification of small-molecule byproducts, which is a challenging and labour-intense type of MS-analysis, rarely performed early in drug development for a broad set of candidates.

Our methods enable the identification of intact ADC, as well as transformations such as deconjugation, maleimide hydrolysis, linker/payload cleavage, or linker/payload modification. It is important to note that, even though a low deconjugation rate is desirable, understanding further alterations, such as payload modifications or linker cleavage, can provide deeper insights into the drug pharmacokinetics and safety profile.

A notable drawback of using maleimide for thiol conjugation is the risk of deconjugation due to the retro-Michael reaction, which can lead to premature drug release in the plasma and potential toxicity [36]. To enhance ADC stability and reduce off-target toxicities, an alternative like bromoacetamide can be employed.

To investigate the linker stability, Her-maleimide and Her-bromoacetamide ADCs were incubated in mouse serum under biological conditions for 3 days. As expected, the transformations detected at 72 h in Her-maleimide were deconjugation (8%) and extensive maleimide hydrolysis (>70%) (Figure 2A,B), whilst Her-bromoacetamide ADC presented a minimal deconjugation (<0.2%) uniquely in the light chain with no detectable hydrolysed species (Figure 2C,D). Hence, these results confirmed that the bromoacetamide linker offered superior stability to the ADC compared to maleimide linkers. Next, we sought to investigate whether these findings were equivalent to those in vivo.

Based on the in vitro data produced, theoretically, both compounds were anticipated to present high stability and potency in vivo. However, the Her-bromoacetamide was expected to be the more stable candidate, exhibiting low off-target effects whilst retaining high potency. As shown in Figure 3A, both ADCs remained notably stable in the mouse model. Interestingly, Her-bromoacetamide exhibited less than 3% of loss over a period of three weeks; however, it presented a more favourable PK profile with slower clearance in mice (Figure 3B) compared to Her-maleimide (Figure 3C).

It is noteworthy that the in vivo results are in accordance with the in vitro findings, i.e., the Her-bromoacetamide suffered less DAR loss across the time course compared to Her-maleimide. It is important to focus on the observed trends rather the DAR loss values, since in vivo and in vitro assays use different workflows and have, in turn, different detection limits. Nonetheless, our internal evaluation of hundreds of ADCs across multiple projects has demonstrated a strong correlation between in vitro stability profiles and corresponding in vivo data; however, specific molecules and corresponding associated data cannot be disclosed, due to intellectual property considerations. This alignment highlights the utility of the current analytical approach for candidate selection early in development. However, as total ADC clearance remains a decisive factor for drug performance, there is a clear need for in vitro assays capable of reliably predicting in vivo clearance. Expanding our analytical toolbox in this direction will further refine and strengthen our preclinical evaluation strategies.

#### 3.2.2. Power BI Integration in Analysis of Serum Stability Datasets

Early drug development is a fast-paced process in which candidate molecules progress from initial design through optimisation via in vitro and preliminary in vivo studies, before entering formal preclinical development. Hence, it is critical to generate or maintain an analytical toolbox able to deliver robust results efficiently [37].

Serum stability and other bioanalytical workflows often require complex, multi-step procedures that generate large datasets. For instance, our current workflow involves several samples (including replicas, time points, and quality controls) that produce several LC-MS files and, therefore, thousands of ions, which, after deconvolution, result in hundreds of masses per ADC candidate. Given the large data volume and the need for timely results, we sought to apply Power BI into our analysis pipeline to increase speed in data processing and visualisation.

Our Power BI pipeline enables rapid analysis of DAR values for the ADC and their individual chains, as well as the more relevant transformations such as deconjugation, or maleimide hydrolysis. In addition, PowerBI can potentially generate a standardised report dashboard, a feature that speeds up the deliveries, helping in the candidate selection.

This implementation has provided significant advantages, such as the following:-Reduced analysis time by enabling fast and automated processing of the deconvoluted MS datasets.-Standardised reports eliminating or reducing manual effort and variation.-Enhanced throughput providing stability data to multiple projects and, in turn, speeding up the candidate selections.-The flexibility in data tailoring, allowing analysis of non-conventional ADCs.

In summary, the use of Power BI enhanced the analysis speed and consistency of complex datasets whilst improving the consistency and quality of the reports.

## 4. Discussion

Comprehensive and robust analytical workflows are essential for effective candidate selection in antibody–drug conjugate development. Early analytics not only guide rational molecular design but are also critical for selection of candidates with the highest likelihood of clinical success. Our integrated workflow, combining enzymatic cleavage analysis, serum stability studies, in vitro cytotoxicity assays, and in vivo pharmacokinetic profiling, has demonstrated substantial value in triaging ADC candidates.

The systematic integration of these bioanalytical methods enabled us to internally increase our yearly productivity 5-fold, allowing the evaluation of ADCs with diverse payloads and mechanisms of action across various serum species. This high-throughput and comprehensive analytic platform may pave the way to accelerate drug development, bringing to the table potent and stable therapeutics with favourable safety profiles. Importantly, our workflow was improved by using automated data analysis and visualisation, reducing turnaround times, increasing throughput, and ensuring consistency in data interpretation across multiple projects and operators.

Our results showed that ADCs containing stable linker handles such as bromoacetamide are promising alternatives to ADCs with conventional maleimide linkers, exhibiting strong resistance to deconjugation. In this regard, the bromoacetamide linker eliminated the retro-Michael reaction. Moreover, ADC manufacturing, characterisation, and cytotoxicity remained comparable among linkers. Notably, in vitro serum stability analysis showed that deconjugation rates dropped from 8% to <0.2% with the bromoacetamide linker compared to maleimide-containing linkers. Importantly, these in vitro results aligned with the PK data, where bromoacetamide presented a lower DAR loss compared to the conventional maleimide-containing linker. The noticeable improvement in stability, both in vitro and in vivo, suggests that bromoacetamide linkers could translate into enhanced therapeutic response and potentially reduced adverse effects. However, validation of these advantages will require subsequent in vivo efficacy studies in tumour-models and toxicity assessments such as dose-range finding (DRF) and rat toxicity studies (RTSs).

The correlation observed between in vitro analytical data and in vivo pharmacokinetic results highlights the value of early analytical screening during candidate selection. Although the statistical power of this study is limited by the sample size, our focus in early drug discovery is on (1) quickly and efficiently screening candidates triaging the most promising compounds and (2) obtaining a descriptive understanding of the ADC PK properties, rather than conducting statistical comparisons. Nevertheless, we also recognise several limitations, most notably, the absence of an in vitro assay for predicting clearance of total ADC. Further development in this area would strengthen our ability to anticipate clinical performance, optimise linker-payload selection, and reduce reliance on animal models. In addition, other important challenges should be assessed, including the reliance on preclinical models and the need for further validation across a broader range of biotherapeutic modalities, such as antibody–protein conjugates and V_H_H-based products. Future studies should incorporate more extensive PK/PD modelling and include clinical samples to further improve translational predictability.

Overall, our work demonstrates that a well-validated, integrated analytical workflow is a powerful tool for streamlining ADC candidate selection progression, improving efficiency, and facilitating the development of safer and more effective cancer therapies.

## 5. Conclusions

The combination of serum stability, cytotoxicity, and PK data creates a strong foundation for candidate selection, and when supplemented with enzymatic analytical results from additional compounds, enables a more comprehensive understanding of the ADC performance and behaviour. As evidenced by our findings, our analytical workflow demonstrates that ADCs with novel linkers such as bromoacetamide achieve enhanced stability and prolonged pharmacokinetics in early development settings, whilst maintaining high cytotoxic efficacy.

These results validate the critical importance of linker and conjugation chemistry optimisation in ADC development and highlight the benefits of extensive in vitro and in vivo profiling using integrated analytical and data visualisation methodologies. To illustrate the benefits of integrating PowerBI into our analysis pipeline, we highlighted that our analysis timelines were reduced by 3-fold and our sample capacity significantly increased with consistent output across analysts. We recognise that other platforms, such as R or Python, could similarly be used to automate analytical processes and achieve comparable improvements. However, Power BI allowed us to implement an integrated workflow with minimal coding expertise, making it suitable for our team and project needs.

The workflow and strategy presented herein support candidate selection with better clinical potential and provide an approach for efficient advancement of ADC therapies in cancer. Overall, our findings provide an example for efficient advancement and prioritisation of ADC therapies.

## Figures and Tables

**Figure 1 cancers-18-00164-f001:**
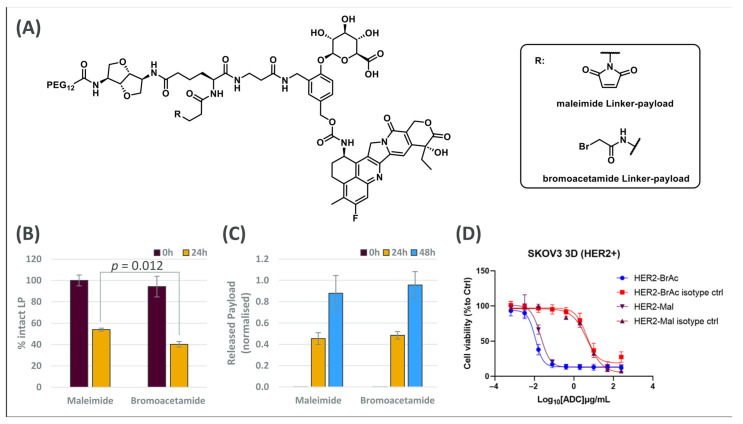
(**A**) Linker-payload structures for the ADCs tested. Both LPs were conjugated to Herceptin-WT producing Her-maleimide and Her-bromoacetamide. (**B**) Enzymatic cleavage of LPs with β-glucuronidase. (**C**) ADC catabolic study in lysosomes or payload release study from both ADCs following incubation in lysosomal extract. (**D**) IC_50_ curves for Her-maleimide (Her2-Mal) and Her-bromoacetamide (Her2-BrAc) alongside the ones obtained from the corresponding isotypes in the SKOV3 cell line cultured as spheroids. (**B**–**D**) Results are shown as mean ± SD (n = 2–4). (**B**,**C**) Statistical analysis was performed for each time point using an unpaired two-tailed *t*-test.

**Figure 2 cancers-18-00164-f002:**
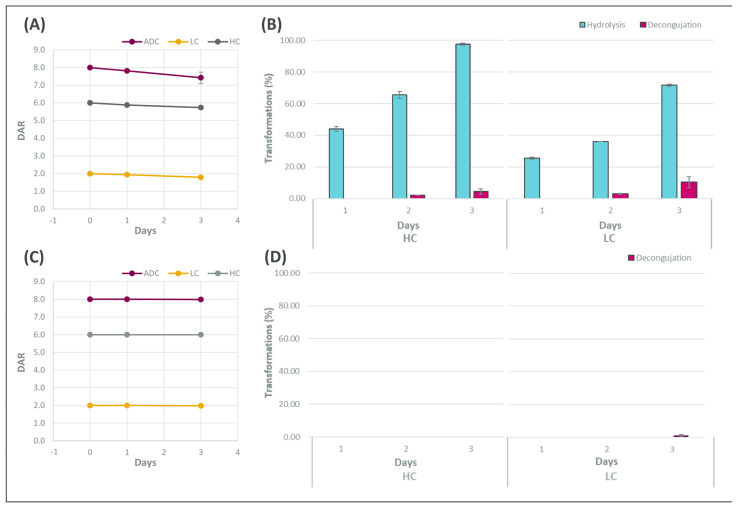
(**A**–**D**) DAR values and biotransformation for Her-maleimide (**A**,**B**) and bromoacetamide (**C**,**D**). DAR plots (**A**,**C**) show the drug-to-ratio per chain, light and heavy (LC and HC), and per ADC, whilst the biotransformation plot (**B**,**D**) presents the typical changes per chain explained previously. (**A**–**D**) Results are shown as mean ± SD (n = 2).

**Figure 3 cancers-18-00164-f003:**
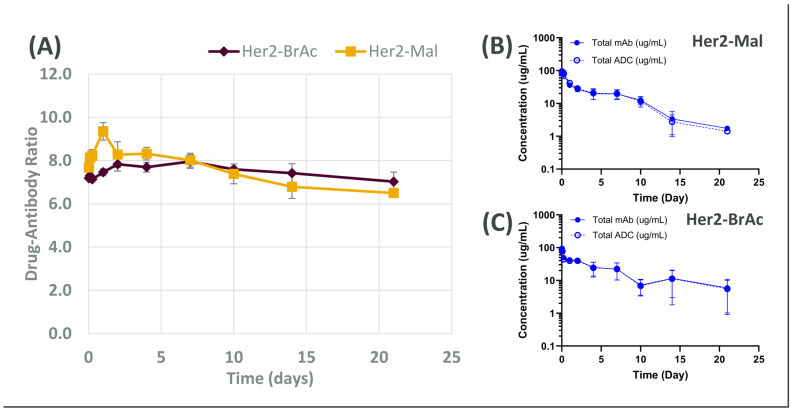
Abbreviations: Her2-Mal, Her-maleimide; Her2-BrAc, Her-bromoacetamide. (**A**) DAR profile of the tested ADCs in mice, presented as mean ± SEM (n = 3). (**B**,**C**) Plasma concentrations of total ADC (exatecan detected) and total mAb (Herceptin detected) at different time points in mice. Results are shown as mean ± SEM (n = 6).

## Data Availability

Data is contained within the article or Appendix A.

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
