# Peer review of "Integrating In Vitro Analytics for Improved Antibody–Drug Conjugate Candidate Selection"

_cancers, 2026, doi:10.3390/cancers18010164_

Round 1
Reviewer 1 Report
Comments and Suggestions for Authors
Dear Authors,
During the review of the manuscript, the following points were noted. Kindly address these comments to enhance the clarity and scientific rigor of the manuscript:
- Abstract:
- Regarding the sentence “Integrating these in vitro assays with powerful data analysis software accelerates structure activity relationship assessments and the identification of stable compounds”: Please clarify how in-vitro data and SAR assessments contribute to identifying stable compounds. In my understanding, in-vitro studies and SAR analyses primarily help identify potent compounds, not necessarily stable ones. Kindly revise or rephrase the sentence for accuracy. (Lines 36 - 38).
- Key-words are too long. Make it short.
- The manuscript includes a Simple Summary section before the abstract. Is this section necessary, given that MDPI already uses a structured abstract format? Please clarify how the Simple Summary differs from the Abstract and whether its inclusion is required.
- Introduction: The introduction is too general and lacks depth. It does not adequately reference previous research in this area. Please include recent studies conducted by other researchers to strengthen the context and relevance of the work.
- Materials and Methodology: Please specify the purification method used for the ADC.
- Please provide the Animal Ethical Committee clearance number for the study.
- Please provide link for the supplementary material.
- ADC based therapy is not a new concept. Kindly justify novelty of your work.
Author Response
1. Abstract:
Regarding the sentence “Integrating these in vitro assays with powerful data analysis software accelerates structure activity relationship assessments and the identification of stable compounds”: Please clarify how in-vitro data and SAR assessments contribute to identifying stable compounds. In my understanding, in-vitro studies and SAR analyses primarily help identify potent compounds, not necessarily stable ones. Kindly revise or rephrase the sentence for accuracy. (Lines 36 - 38).
We thank the reviewer for highlighting this important point.
As stated in the abstract, our approach involves advanced analytical methods to triage ADCs. We can determine which compound structures are more cleavable within cells and which are more stable in plasma across multiple species (human, rat, mouse) by combining enzymatic assays with plasma stability assays. To avoid confusion with the inherent stability of the compound, we have revised the sentence to specify that the stability refers specifically to plasma stability.
The revised sentence now reads:
“Integrating these in vitro assays with powerful data analysis software accelerates structure activity relationship assessments and the identification of stable compounds in plasma.”
We thank again the reviewer for the helpful feedback.
2. Key-words are too long. Make it short.
Thank you for your feedback about the length of the keywords.
Now, keywords are as follows:
1; Antibody-drug conjugates 2; Enzymatic assay 3; ADC plasma stability 4; Computational integration
3. The manuscript includes a Simple Summary section before the abstract. Is this section necessary, given that MDPI already uses a structured abstract format? Please clarify how the Simple Summary differs from the Abstract and whether its inclusion is required.
We thank the reviewer for his/her comment regarding the Simple Summary section.
This section is a specific requirement of the journal. According to the journal’s guidelines: summary should be written in one paragraph before the Abstract in layman’s terms, to explain why the research is being suggested, what the authors aim to achieve, and how the findings from this research may impact the research community. Submissions without a simple summary will be returned directly.
Therefore, we have provided this section in accordance with the journal’s instructions.
4. Introduction: The introduction is too general and lacks depth. It does not adequately reference previous research in this area. Please include recent studies conducted by other researchers to strengthen the context and relevance of the work.
Thank you for highlighting the need for greater depth and additional references in the introduction.
We regret to have missed some relevant work in the field. In response, we have reformatted the introduction and included references to recent studies by other researchers that we have considered appropriate. We hope these additions strengthen the context and relevance of the work, as the reviewer suggested.
5. Materials and Methodology: Please specify the purification method used for the ADC.
Thank you for your suggestion. The purification method has now been added immediately after the conjugation method in the Materials and Methods section.
“Briefly, both linker-payloads were added as a DMSO solution (16 molar equiva-lent/antibody, 0.64 µmol, in 0.13mL DMSO) to Trastuzumab antibody solution in PBS, 1 mM EDTA, pH 7.4 (6.0 mg, 40 nmol) for a 10% (v/v) final DMSO concentration. For bro-moacetamide linker conjugation, 15 % of borate buffer pH 8.5 was added to a final pH 8.3. Both linker solutions were incubated at room temperature for 1 hour with gentle shaking and quenched by addition of N-acetyl cysteine (3.2 µmol, 32.0 mL at 100 mM), followed by purification in PBS pH 7.4 using AKTA-SECPrep (HiLoad® 26/600 Superdex® 200 pg, GE28-9893-36, Cytiva) and formulated in 20 mM Histidine/Histidine HCl, 240 mM sucrose pH 6.0 by spin filtration using a 15 mL Amicon Ultracell 30 kDa MWCO spin filter (UFC5030, Millipore), sterile-filtered and analysed.”
6. Please provide the Animal Ethical Committee clearance number for the study.
Thank you for your request regarding the ethical approval for the animal experiments.
The ethical approval number for the animal experiment is AUP-22-34.
7. Please provide link for the supplementary material.
Thank you for your request about the link to the supplementary materials.
I have checked with the editor and the link to the supplementary files is generated and published by the journal after the article is accepted and online. Authors do not need to provide a pre-publication link.
However, I have ensured that all supplementary documents have been provided with the replies of the reviewer’s comment in case they were not previously available to the reviewer.
8. ADC based therapy is not a new concept. Kindly justify novelty of your work.
We thank the reviewer for the comment on the novelty of ADC-based therapy. We regret the novel aspects of our work were not sufficiently clear and appreciate the opportunity to further clarify our contribution.
The novelty of this manuscript lies in our analytical toolbox, which integrates plasma, lysosomal, and enzymatic stability assays with advanced data analysis software to provide comprehensive and predictive assessment of ADC candidates. In early drug development, the speed in deciding which of the hundreds of candidates would proceed to animal studies is critical, given the time and resources needed for in vivo testing. Our workflow and computational integration enable efficient triage, allowing us to quickly identify the best candidates for further evaluation. Finally, I would like to mention that we observed strong correlation between in vitro plasma stability and in vivo PK datasets across more than 100 ADCS from different projects (unfortunately, the correlation details cannot be disclosed due to intellectual property considerations), supporting the predictive strength of our early-stage screening strategy.

Reviewer 2 Report
Comments and Suggestions for Authors
This manuscript studies integrated approach for the early-stage selection of Antibody-Drug Conjugate (ADC) candidates. The authors combine enzymatic assays, serum stability studies, in vitro cytotoxicity, and in vivo pharmacokinetics to evaluate ADC properties, with a focus on linker chemistry (maleimide vs. bromoacetamide). A key strength is the incorporation of Power BI for data analysis and visualization, which enhances consistency of results. The study demonstrates a correlation between in vitro stability data and in vivo performance, supporting the use of such approach for candidate evaluation.
However, after examining the manuscript I have following questions and suggestions:
1) The manuscript presents quantitative data on Figures as mean ± SD or SEM but does not report any statistical tests to support claims of similarity or difference between the Her-maleimide and Her-bromoacetamide ADCs. Please, perform needed statistical tests (unpaired t-test, one-way or two-way ANOVA) on all comparative data. Report obtained p-values in the figures legends.
2) The in vivo pharmacokinetic study uses a small sample size (n=3 for DAR, n=6 for concentration). With such small n, the study may be weak to detect biologically relevant differences in PK parameters like clearance. Please, include a brief justification for the chosen sample size. Acknowledge the limited power of the study as a limitation in the discussion.
3) The description of the Power BI integration is a key novelty but it is not deep described. The statement that analysis timelines were "reduced by 3-fold" is a good, but the manuscript lacks details on how this was achieved, making it difficult for others to reproduce. Please, expand the Methods section or include a supplementary section with more technical details.
4) The manuscript states that "our evaluation of hundreds of ADCs has shown a strong correlation between in vitro stability profiles and the corresponding in vivo data." This is a strong claim, but is not supported by any data, regression analysis, or correlation coefficients within the manuscript. Please add.
5) The conclusion that "bromoacetamide linkers could translate into enhanced therapeutic response and potentially reduced adverse effects" is speculative based on the data presented. The study demonstrates improved stability and PK, but does not include in vivo efficacy or toxicity data in tumor-bearing models to directly support this claim. Please, rephrase.
6) The use of "isotypic ADC analogues" as controls in the cytotoxicity assay (Figure 1) is mentioned but not clearly defined. It is unclear what antibody was used for the isotype control. Please, clearly identify the isotype control antibody in the Methods section.
7) Figure 3A. The y-axis label "Drug Antibody Ratio" should be "Drug-to-Antibody Ratio". Also for all figures where error bars are present, the nature of the error (SD vs. SEM) and the sample size (n) should be explicitly stated in the figure legends, not only in the results text.
8) The authors correctly identify the absence of an in vitro assay for predicting total ADC clearance as a limitation. While developing such an assay may be beyond the scope of this paper, the Discussion should briefly propose potential experimental avenues or models for future work.
Finally, manuscript presents well-executed study with clear relevance to the field. It merits publication after major revisions, to address statistical reporting, sample size justification, and clarification of speculative claims.
Author Response
Reviewer2
1) The manuscript presents quantitative data on Figures as mean ± SD or SEM but does not report any statistical tests to support claims of similarity or difference between the Her-maleimide and Her-bromoacetamide ADCs. Please, perform needed statistical tests (unpaired t-test, one-way or two-way ANOVA) on all comparative data. Report obtained p-values in the figures legends.
Thank you for your suggestion regarding the reporting of statistical tests for claims of similarity or difference between the Her-maleimide and Her-bromoacetamide ADCs. As recommended, we have performed unpaired t-tests for the enzymatic and catabolic studies. Details of the statistical analysis are now specified in the Materials and Methods section. Additionally, the only significant p-value has been included in the relevant figure (Figure 1).
For the other analyses, and as outlined in the manuscript, our analytical approach was guided by the following objectives:
- For the in vitro data, our goal is to identify molecules that meet predefined criteria across cytotoxicity and plasma stability assays triaging, in turn, compounds as the first cutoff in drug development. Although we are presenting results for two ADCs here, our focus is not on comparing differences between them; rather, we apply a go/no-go decisions. Both ADCs met the required standards and were selected to proceed to animal studies. Based on these initial data, each compound will progress to rat toxicity studies and, if they continue to meet the required benchmarks, to further preclinical evaluation.
- For the PK data, our focus was on obtaining a descriptive understanding of each molecule PK properties rather than conducting statistically powered analyses.
As a general statement, we have clarified in the “Data Analysis and Visualisation” section that “Statistical analysis was performed only for enzymatic and catabolic studies. Cutoff criteria were used for in vitro cytotoxicity and plasma stability datasets as part of a go/no-go decision process whilst the objective for in vivo study was to obtain a descriptive overview of PK properties of the molecule.”
Thank you again for your helpful feedback.
2) The in vivo pharmacokinetic study uses a small sample size (n=3 for DAR, n=6 for concentration). With such small n, the study may be weak to detect biologically relevant differences in PK parameters like clearance. Please, include a brief justification for the chosen sample size. Acknowledge the limited power of the study as a limitation in the discussion.
Thank you for your comment regarding the sample size used in the in vivo pharmacokinetic study.
As noted, we performed naïve-pooled analysis with sparse sampling, aiming to obtain a descriptive understanding of the molecule PK properties rather than to conduct statistically powered analyses. This approach aligns with our focus on rapid candidate triage during early drug discovery.
We have revised the discussion to acknowledge this limitation as follows:
“The correlation observed between in vitro analytical data and in vivo pharmacokinetic results highlights the value of early analytical screening during candidate selection. Although the statistical power of this study is limited by the sample size, our focus in early drug discovery is on (1) quickly and efficiently screening candidates triaging the most promising compounds and (2) obtaining a descriptive understanding of the ADC PK properties, rather than conducting statistical comparisons.”
3) The description of the Power BI integration is a key novelty but it is not deep described. The statement that analysis timelines were "reduced by 3-fold" is a good, but the manuscript lacks details on how this was achieved, making it difficult for others to reproduce. Please, expand the Methods section or include a supplementary section with more technical details.
Thank you for highlighting the importance of a detailed description of the Power BI integration in our workflow.
In response, we have expanded the Methods section and added relevant Power BI code in the supplementary material, located immediately after the ADC characterization details.
We hope this provides the necessary technical information to support reproducibility.
4) The manuscript states that "our evaluation of hundreds of ADCs has shown a strong correlation between in vitro stability profiles and the corresponding in vivo data." This is a strong claim, but is not supported by any data, regression analysis, or correlation coefficients within the manuscript. Please add.
Thank you for your comment regarding the support for our statement on the correlation between in vitro stability profiles and in vivo data.
Unfortunately, due to intellectual property restrictions, we are unable to share additional details. However, I can confirm that the linear correlation between the DAR loss at 96h (serum stability data) and PK AUC ratio total ADC/Total mAb (final point for in vivo PK data) across more than 100 ADCs from different projects, with different Abs and linkerpayloads, presented an R² value of 0.693.
However, we have expanded the statement about this correlation to clarify and support our claim.
“Nonetheless, our internal evaluation of hundreds of ADCs across multiple projects has demonstrated a strong correlation between in vitro stability profiles and corresponding in vivo data; however, specific molecules and associated data cannot be disclosed due to intellectual property considerations”
Thank you again for highlighting this important point.
5) The conclusion that "bromoacetamide linkers could translate into enhanced therapeutic response and potentially reduced adverse effects" is speculative based on the data presented. The study demonstrates improved stability and PK, but does not include in vivo efficacy or toxicity data in tumor-bearing models to directly support this claim. Please, rephrase.
Thank you for your comment about the conclusion on bromoacetamide linkers.
We agree that, based on the available data, this statement is speculative and have accordingly expanded the sentence for clarity. The revised conclusion now includes the follow:
“However, validation of these advantages will require subsequent in vivo efficacy studies in tumour-models and toxicity assessments such as dose-range finding (DRF) and rat toxicity studies (RTS).”
6) The use of "isotypic ADC analogues" as controls in the cytotoxicity assay (Figure 1) is mentioned but not clearly defined. It is unclear what antibody was used for the isotype control. Please, clearly identify the isotype control antibody in the Methods section.
Thank you for your observation about the lack of information on the isotype ADC analogues.
Information about the isotype control antibody has been added to the Methods section, at the end of the “In Vitro Cytotoxicity Assays” section. The following clarification has been included:
“ADC controls used in the studies were the isotype ADC analogues, NIP228-WT.”
Thank you again for your helpful suggestion.
7) Figure 3A. The y-axis label "Drug Antibody Ratio" should be "Drug-to-Antibody Ratio". Also for all figures where error bars are present, the nature of the error (SD vs. SEM) and the sample size (n) should be explicitly stated in the figure legends, not only in the results text.
Thank you for your suggestions regarding Figure 3A and the figure legends. The y-axis label in Figure 3A has been corrected to “Drug-to-Antibody Ratio”.
About the nature of the error bars, we have reviewed the figure legends and ensured that the type of error (SD or SEM) and sample size (n) are specified for all figures as required.
Thank you again for highlighting these mistakes.
8) The authors correctly identify the absence of an in vitro assay for predicting total ADC clearance as a limitation. While developing such an assay may be beyond the scope of this paper, the Discussion should briefly propose potential experimental avenues or models for future work.
Thank you for your suggestion about future experimental approaches for predicting total ADC clearance.
We are currently developing few in vitro assays for ADC clearance which could potentially be used to predict in vivo outcomes through successful correlation with our existing in vivo dataset. However, we aim to patent any method that reliably predicts in vivo ADC clearance. Thus, due to intellectual property considerations, we cannot disclose further details beyond what is currently described in the manuscript.
We appreciate your understanding and thank you for raising this important point.
Reviewer 3 Report
Comments and Suggestions for Authors
The manuscript lacks novelty. The methods described in the manuscript are well-known and have been reported/published.
The manuscript is also not well prepared.
- Instead of dumping all data into the manuscript, the results should be presented properly. For example, the supplementary figures are representative characterization profiles of the ADCs. Though these characterizations are routine in the ADC field, for readers who are not familiar with ADCs, what are the peak assignments, what are the details of the methods and materials? And most importantly, why are these figures necessary for this manuscript?
- The results are not well interpreted or discussed. For example, in Figure 3, it is weird to emphasize focusing on a trend that looks like no difference. How the “trend” is defined. It is pointed out in lines 345-346 that there are data of hundreds of ADCs with minimal variation, yet a specific example is selected in the manuscript that only shows a “trend”. In addition, what are the clearance values mentioned in the manuscript, as well as other PK parameters? It is important to provide actual numbers other than “eyeballing”. And also, how do these observations reflect in the endpoint readouts, e.g., efficacy at least in animal models? Does a faster clearance necessarily translate into a different efficacy outcome?
- It is also emphasized in the manuscript that Power BI was integrated into the data analysis pipeline. It would be beneficial for the readers if details were provided. What is the setup, validations? Why is Power BI used instead of many other commercial or customized AI (or non-AI) tools?
Author Response
1) Instead of dumping all data into the manuscript, the results should be presented properly. For example, the supplementary figures are representative characterization profiles of the ADCs. Though these characterizations are routine in the ADC field, for readers who are not familiar with ADCs, what are the peak assignments, what are the details of the methods and materials? And most importantly, why are these figures necessary for this manuscript?
Thank you for your suggestions regarding the presentation of results and supporting figures. The supporting information has been updated to include peak assignments to help in species identification. In addition, details of the purification method have now been added under the bioconjugation of ADCs.
The characterisation figures are provided to demonstrate that both ADCs analysed in this study were properly characterised. We consider that these figures are necessary for transparency, confirming the validity and reliability of our analytical findings. Finally, we would like to highlight that the main scope of this manuscript is the development of integrated analytical workflow for the early-stage assessment and triage of ADC candidates.
Thank you again for highlighting these points.
2) The results are not well interpreted or discussed. For example, in Figure 3, it is weird to emphasize focusing on a trend that looks like no difference. How the “trend” is defined. It is pointed out in lines 345-346 that there are data of hundreds of ADCs with minimal variation, yet a specific example is selected in the manuscript that only shows a “trend”. In addition, what are the clearance values mentioned in the manuscript, as well as other PK parameters? It is important to provide actual numbers other than “eyeballing”. And also, how do these observations reflect in the endpoint readouts, e.g., efficacy at least in animal models? Does a faster clearance necessarily translate into a different efficacy outcome?
Thank you for your detailed comments and suggestions regarding the interpretation and discussion of our results. We appreciate the reviewer’s insights into the presentation of trends and the request for additional PK data and efficacy endpoints.
The primary aim of this manuscript is to contribute to the ADC expert community by focusing on the analytical workflows and stability screening strategies relevant for early-stage development, rather than providing comprehensive PK or efficacy comparisons across a wider spectrum of compounds. While our dataset involves hundreds of ADCs from different projects, the figures and trends presented are intended to illustrate typical observations and the practical application of our integrative screening approach. Given the manuscript’s scope, we have stressed the workflow design, decision criteria, and the validation of key analytical steps, rather than detailed statistical comparisons or efficacy outcomes, which would require extensive in vivo studies outside the scope of this work and early-discovery development.
We agree that in-depth PK data and efficacy endpoints are important for a broader scientific audience; however, the current work is specifically tailored to the needs and interests of ADC experts. Expanding the discussion to include comprehensive PK metrics and animal model endpoints would move beyond the intended focus. We respectfully suggest that these topics, while valuable, are best addressed in future studies designed to evaluate in vivo efficacy and detailed PK relationships.
Thank you again for your thoughtful feedback.
3) It is also emphasized in the manuscript that Power BI was integrated into the data analysis pipeline. It would be beneficial for the readers if details were provided. What is the setup, validations? Why is Power BI used instead of many other commercial or customized AI (or non-AI) tools?
Thank you for your constructive feedback regarding the integration of Power BI in our data analysis pipeline. Details of the Power BI coding have been provided in the Supplementary Information.
We would like to clarify that our intention was not to promote Power BI as the definitive tool, but rather to describe our practical workflow. Initially, we used R within our analysis pipeline; however, this approach required us to have advanced coding skills. Power BI was then chosen due to its accessibility and ease use for team members without programming expertise, thereby streamlining our workflow.
Overall, our main recommendation was the implementation of an integrated analysis pipeline, regardless of the specific software platform used. We have now clarified this point in the manuscript.
“To illustrate the benefits of integrating PowerBI into our analysis pipeline, we highlighted that our analysis timelines were reduced by 3-fold and our sample capacity significantly increased with consistent output across analysts. We recognise that other platforms, such as R or Python, could similarly be used to automate analytical processes and achieve comparable improvements. However, Power BI allowed us to implement an integrated workflow with minimal coding expertise, making it suitable for our team and project needs.”
Reviewer 4 Report
Comments and Suggestions for Authors
The manuscript presents an integrated in vitro analytical workflow to rapidly assess the stability and efficacy of ADC candidates, for easier and faster ADC screening. The results show in vitro-in vivo correlations for the two ADCs studied. The article can be accepted after minor revision:
- Line 344: The authors stated that “Nonetheless, our evaluation of hundreds of ADCs has shown a strong correlation between in vitro stability profiles and the corresponding in vivo data, with minimal variation.” Proper bibliography should be included to confirm this statement, as only two ADCs were studied in the present article.
- Even though the article states that no statistical analysis was performed (line 192), I wonder why the authors chose not to perform statistical tests.
Author Response
1) Line 344: The authors stated that “Nonetheless, our evaluation of hundreds of ADCs has shown a strong correlation between in vitro stability profiles and the corresponding in vivo data, with minimal variation.” Proper bibliography should be included to confirm this statement, as only two ADCs were studied in the present article.
Rephrase the sentence.
Thank you for highlighting the need to clarify this statement. The sentence was based on internal data obtained from in vitro plasma stability assays and in vivo PK studies performed on a large number of ADCs across multiple projects. We acknowledge that the previous wording may have caused confusion, so we have expanded the sentence for clarity:
“Nonetheless, our internal evaluation of hundreds of ADCs across multiple projects has demonstrated a strong correlation between in vitro stability profiles and corresponding in vivo data; however, specific molecules and associated data cannot be disclosed due to intellectual property considerations.”
We appreciate your suggestion and have amended the manuscript accordingly.
2) Even though the article states that no statistical analysis was performed (line 192), I wonder why the authors chose not to perform statistical tests.
Thank you for your question regarding the absence of statistical analysis. We have performed unpaired t-tests for the enzymatic and catabolic studies. Details of the statistical analysis are now specified in the Materials and Methods section. Additionally, the only significant p-value has been included in the relevant figure.
For the other analyses, as outlined in the manuscript, our analytical approach was guided by the following objectives:
- For the in vitro data, our goal is to identify molecules that meet predefined criteria across cytotoxicity and plasma stability assays triaging, in turn, compounds as the first cutoff in drug development. Although we are presenting results for two ADCs here, our focus is not on comparing differences between them; rather, we apply a go/no-go decisions. Both ADCs met the required standards and were selected to proceed to animal studies. Based on these initial data, each compound will progress to rat toxicity studies and, if they continue to meet the required benchmarks, to further preclinical evaluation.
- For the PK data, our focus was on obtaining a descriptive understanding of each molecule PK properties rather than conducting statistically powered analyses.
We have further clarified in the manuscript when statistical analysis has been performed contained in the “Data Analysis and Visualisation” section. The revised sentence now reads:
“Statistical analysis was performed only for enzymatic and catabolic studies. Cutoff criteria were used for in vitro cytotoxicity and plasma stability datasets as part of a go/no-go decision process whilst the objective for in vivo study was to obtain a descriptive overview of PK properties of the molecule.”
Thank you again for your attention to this aspect of our methodology.
Round 2
Reviewer 2 Report
Comments and Suggestions for Authors
The authors have provided a comprehensive revision of their manuscript, directly addressing the majority of my critical comments. The responses are clear, and the corresponding changes to the manuscript text are appropriate and effectively strengthen the scientific rigor and clarity of the work. The study presents a valuable, well-executed work for ADC development with clear practical significance for the field.
The manuscript has been improved and is now suitable for publication. I recommend acceptance. No further scientific revisions are required.